The microbiota of Drosophila suzukii influences the larval development of Drosophila melanogaster

Solomon Gabrielle M.
Dodangoda Hiruni
McCarthy-Walker Tylea
Ntim-Gyakari Rita
Newell Peter D. peter.newell@oswego.edu
Department of Biological Sciences, State University of New York at Oswego , Oswego , NY , United States of America
Mikheyev Alexander
Electronic publication date: 2019 Nov 19
Publication date: 2019
Volume: 7
Electronic Location ID: e8097
Received 2019 Aug 13; Accepted 2019 Oct 24
Copyright: ©2019 Solomon et al.
Copyright year: 2019
Copyright holder: Solomon et al.
License: This is an open access article distributed under the terms of the Creative Commons Attribution License, which permits unrestricted use, distribution, reproduction and adaptation in any medium and for any purpose provided that it is properly attributed. For attribution, the original author(s), title, publication source (PeerJ) and either DOI or URL of the article must be cited.
License URL: https://creativecommons.org/licenses/by/4.0/

Keywords: Symbiosis, Spotted Wing Drosophila, Larval development, Yeast, Gut microbiota

Funding: Rice Creek Associates small grants program Possibility Scholarship Office of the Provost of the State University of New York at Oswego This work was supported by the Rice Creek Associates small grants program, a Possibility Scholarship awarded to Hiruni Dodangoda, and a grant for scholarly and creative activity from the Office of the Provost of the State University of New York at Oswego awarded to Gabrielle Solomon and Peter Newell. The funders had no role in study design, data collection and analysis, decision to publish, or preparation of the manuscript.

==============================
Microorganisms play a central role in the biology of vinegar flies such as Drosophila suzukii and Drosophila melanogaster: serving as a food source to both adults and larvae, and influencing a range of traits including nutrition, behavior, and development. The niches utilized by the fly species partially overlap, as do the microbiota that sustain them, and interactions among these players may drive the development of crop diseases. To learn more about how the microbiota of one species may affect the other, we isolated and identified microbes from field-caught D. suzukii, and then characterized their effects on D. melanogaster larval development time in the laboratory. We found that the D. suzukii microbiota consistently included both yeasts and bacteria. It was dominated by yeasts of the genus Hanseniaspora, and bacteria from the families Acetobacteraceae and Enterobacteriaceae. Raising D. melanogaster under gnotobiotic conditions with each microbial isolate individually, we found that some bacteria promoted larval development relative to axenic conditions, but most did not have a significant effect. In contrast, nearly all the yeasts tested significantly accelerated larval development. The one exception was Starmerella bacillaris, which had the opposite effect: significantly slowing larval developmental rate. We investigated the basis for this effect by examining whether S. bacillaris cells could sustain larval growth, and measuring the survival of S. bacillaris and other yeasts in the larval gut. Our results suggest S. bacillaris is not digested by D. melanogaster and therefore cannot serve as a source of nutrition. These findings have interesting implications for possible interactions between the two Drosophilia species and their microbiota in nature. Overall, we found that microbes isolated from D. suzukii promote D. melanogaster larval development, which is consistent with the model that infestation of fruit by D. suzukii can open up habitat for D. melanogaster. We propose that the microbiome is an important dimension of the ecological interactions between Drosophila species.

Introduction

Microorganisms are an integral part of animal biology (McFall-Ngai et al., 2013). This is especially true for Drosophila species, whose associated microbes are known to affect nutrition, immunity, and a range of other traits (Wong, Vanhove & Watnick, 2016; Martino, Ma & Leulier, 2017). As a model organism, Drosophila melanogaster has been the focus of a great deal of research into the mechanisms of host-microbiota interactions (Buchon, Broderick & Lemaitre, 2013; Douglas, 2018). In addition, D. melanogaster has emerged as a useful model for studying ecological interactions that shape the assembly of microbial communities (Adair et al., 2018). The possibility of integrating knowledge across scales, from the molecular to the ecological, makes this a very promising system for these investigations.

Drosophila suzukii is an agricultural pest that infests soft and stone fruit. Since its recent arrival in North America, this invasive species has spread rapidly causing significant economic damage due to crop loss (Walsh et al., 2011; Dos Santos et al., 2017). Unlike its congeneric relatives, D. suzukii lays its eggs in sound, ripening or ripe fruit by means of a serrated ovipositor (Lee et al., 2011). In doing so, the fly introduces microorganisms that hasten the spoilage of the fruit and serve as food for developing larvae (Ioriatti et al., 2015). Through this lifestyle adult D. suzukii can vector microbes that damage fruit crops including the yeasts and acetic acid bacteria (AAB) that cause sour rot, as well as other fungal pathogens (Rombaut et al., 2017; Lewis et al., 2019). The threats posed by this invasive species likely extend beyond those to agriculture because they can utilize fruit from a broad range of plants (Lee et al., 2015; Poyet et al., 2015). As it spreads into new areas, D. suzukii likely impacts the fitness of related species such as D. melanogaster, which was the focus of this study.

The niches of D. melanogaster and D. suzukii partially overlap, as do the taxonomic groups of microorganisms typically associated with each species. Both flies promote the development of sour rot disease in grapes (Barata et al., 2012; Rombaut et al., 2017), and surveys of the microorganisms associated with either Drosophila species have found a number of groups in common including yeasts such as Hanseniaspora uvarum and Pichia kluyveri, and bacteria such as Acetobacter spp. and Gluconobacter spp. (Chandler et al., 2011; Chandler, Eisen & Kopp, 2012; Hamby et al., 2012; Staubach et al., 2013; Vacchini et al., 2017; Bost et al., 2018). Field and lab experiments by Rombaut et al. (2017) found that D. suzukii infestation of grapes promoted the development of sour rot and subsequent utilization of the rotting fruit by larval D. melanogaster. In contrast to this potentially beneficial relationship between the fly species, other studies have suggested that D. melanogaster can outcompete D. suzukii when the two are given access to the same oviposition substrate (Dancau et al., 2017; Shaw et al., 2018). While it is likely that the two Drosophila species and their associated microorganisms are interacting wherever their ranges overlap, much remains to be learned about the nature of these interactions, their broader ecological implications, and how they affect D. suzukii invasion.

The primary goal of this study was to examine the impact of yeasts and bacteria isolated from D. suzukii on D. melanogaster larval development time (between egg deposition and pupariation). D. melanogaster females are attracted to oviposit on fruit that is actively fermenting (Fischer et al., 2017; Rombaut et al., 2017), and the development time of their larvae is a trait influenced by microbiota and relevant to fitness (Broderick & Lemaitre, 2012). We conducted our experiments under gnotobiotic conditions in which individual microbial species were associated with the host to monitor the effect of each isolate independently (Koyle et al., 2016). Prior studies have identified significant genetic and phenotypic differences between bacteria isolated from field-caught flies and those found in laboratory D. melanogaster (Winans et al., 2017; Pais et al., 2018). Our objective was to identify significant interactions between developing Drosophila and new microbial isolates for further study. An additional goal of the study was to monitor the presence of D. suzukii in an understudied location, Oswego County, New York, USA, where no data had been previously reported regarding D. suzukii presence or abundance.

Materials & Methods

Our study consisted of four phases. First, we caught wild Drosophilidae and sampled them for microorganisms. Second, we conducted preliminary taxonomic identifications on a subset of microbial isolates chosen for further study. Third, we measured the larval development time of gnotobiotic D. melanogaster mono-associated with these isolates. Finally, we investigated the basis for the prolonged larval development observed when D. melanogaster was reared with the yeast Starmerella bacillaris. This investigation included a) assessing the ability of larvae to survive on a diet of S. bacillaris and b) assessing the survival of S. bacillaris cells consumed by larvae.

Collection of wild Drosophila

Ten to twelve traps were set and monitored continuously from June 12 to July 31, 2017. Two sites, each about 3 hectares in size, were targeted. One site was centered at Rice Creek Field Station of the State University of New York at Oswego, Oswego, NY (43.430653, −76.549758). This site included both wooded and open areas. The second site was at a nearby commercial fruit orchard where a range of fruit trees and shrubs are cultivated, about 900 m from the first site. Traps were hung one meter off the ground in shaded areas and were mainly located in cherry trees in the orchard. There was no indication of D. suzukii infestation at the orchard before or during the collection period.

Traps were constructed from plastic cups containing holes in the middle, and a plastic cover. Vinegar-dough bait was placed inside a separate sample container within the trap, securely covered with nylon mesh to prevent contact between the flies and the bait. The bait recipe for one trap was 2 g sugar, 0.325 g dry active bread yeast, 17.25 g whole wheat flour, 1 ml apple cider vinegar, and 25 ml water. Monitoring took place about once every four days, and fresh bait was introduced at the same interval. After collection, flies were anesthetized with CO2, and sorted under a dissecting microscope. Male and female D. suzukii and D. melanogaster were identified according to Werner, Steenwinkel & Jaenike (2018) and kept for experimentation.

Selective plating procedure

Flies of interest were placed in a homogenization tube with 125 µl of sterile phosphate-buffered saline (PBS) solution and ∼100 µl of autoclaved ceramic beads (1.4 mm diameter; Mo Bio Cat. # 13113-325). Each fly was homogenized individually for 10 s on high (Biospec Products, model OA60AP-11-1WB). Two dilutions were created (10−1 and 10−2) and 20 µl of each of the three concentrations were spread plated onto two different types of media: GYP media (selective for yeast) contained 20 g/L glucose, 10 g/L peptone, 10 g/L yeast extract, 5 g/L Na-acetate, 12 g/L agar, 980 ml DI water, 0.02 g/L tetracycline, and 0.03 g/L chloramphenicol; BM media (selective for bacteria) contained 10 g/L glucose, 10 ml 50% glycerol, 10 g/L peptone, 5 g/L yeast extract, 15 g/L agar, 980 ml DI water, 10 ml ethanol, 0.01% cyclohexidine, and 0.01 g/L natamycin. Plates were incubated (30° C) for two to three days. Two to three colony types were chosen randomly from each plate and streaked for isolation.

Isolation of DNA from microorganisms

A liquid culture was grown from a single colony in YPD medium containing 10 g/L yeast extract, 10 g/L peptone, and 10 g/L dextrose. Cultures were shaken at 220 rpm at 30° C for 24 h. Promega Wizard Genomic DNA Purification kit was used to isolate DNA from bacteria and yeast according to the instructions.

Identification of microorganisms by PCR and sequencing

PCR targeted the 16S rRNA gene from bacteria (Marchesi et al., 1998) or the rRNA ITS regions from yeast (White et al., 1990). Recipe for one reaction with bacterial DNA template: 29.5 µl PCR water, 10 µl 5x ONETaq Buffer, 2 µl 2 mM DNTP’s, 1.5 µl 20 µM 16S 63F Primer (5′-CAGGCCTAACACATGCAAGTC-3′), 1.5 µl 20 µM 16S 1492R Primer (5′-GGTTACCTTGTTACGACTT-3′), and 0.5 µl Onetaq polymerase (New England Biolabs). Cycling parameters: 60s 95 °C, three times (15s 95 °C, 20s 54 °C, 75s 68 °C), thirty times (15s 95 °C, 20s 58 °C, 75s 68 °C), 5 min 68 °C. Recipe for one reaction of yeast DNA template: 29.5 µl PCR water, 10 µl 5×  ONETaq Buffer, 2 µl 2 mM DNTP’s, 1.5 µl ITS1 primer (5′-TCCGTAGGTGAACCTGCGG-3′), 1.5 µl ITS4 primer (5′-TCCTCCGCTTATTGATATGC-3′), and 0.5 µl Onetaq polymerase. Cycling parameters: 60s 95 °C, thirty times (15s 95 °C, 20s 52 °C, 30s 68 °C), 3 min 68 °C. All primer stock solutions had 20 µM concentration, and templates were adjusted to 2.5 µg per reaction. Gel electrophoresis was used to confirm the presence and purity of PCR products. Products were purified with the GeneJET PCR Purification Kit (Thermo Scientific). Bacterial 16S rRNA gene was sequenced by Genewiz Inc. via automated Sanger sequencing with either the 63F primer (to target variable regions V1–V3, or the 1492R primer to target variable regions V7–V79). Chromatograms were inspected for ambiguous base calls, and raw sequences were trimmed from either end to eliminate them. The longest representative sequence for each isolate was chosen to be used as a BLAST query for the NCBI 16S rRNA database (bacteria) or nr/nt for yeasts using default parameters. Our sequences were deposited with NCBI; the bacterial sequences have accession numbers MN197709 –MN197729, and the yeast sequences MN209205 –MN209223.

Development of gnotobiotic D. melanogaster

Gnotobiotic D. melanogaster were generated and reared as described by Newell & Douglas (2014). Briefly, embryos freshly deposited by Canton S flies (Wolbachia free; obtained from N. Buchon, Cornell University) were collected and dechorionated with 0.6% hypochlorite. After washing thrice with sterile water, 25–40 embryos were aseptically transferred to sterile fly diet (100 g/L brewer’s yeast, 100 g/L dextrose, 12 g/L agar). Microbial cultures were grown in YPD, shaking at 220 rpm, at 30 °C for 24 h. Optical densities of the cultures were measured at 600nm and normalized to OD 0.2 via centrifugation and resuspension in sterile PBS. 50 µl of the desired cell suspension was added directly to each vial. Drosophila was reared at 24.5 °C on a 12 h light, 12 h dark cycle. Larval development was monitored and compared by recording pupariation events three times daily. Development experiments were grouped into five different blocks, each including the axenic treatment as a control. Each microbial treatment was tested in two or three different blocks and compared to the aggregate axenic data as described below.

Measurement of microbial density in Drosophila diet

To estimate microbial cell density in the Drosophila diet after the larval developmental period, microbes were collected from the surface of the food and vial seven days after egg deposition, serially diluted, and spot plated. Five ml of sterile PBS were added to each vial, and the vial was sealed and vortexed on high for eight seconds. Liquid in the vial was sampled and serially diluted to 10−8 in sterile PBS. Five µl aliquots of each dilution were spotted onto YPG agar plates in triplicate. Colonies were counted in spots yielding between 5 and 50 colonies.

Larval survival on whole-yeast diet

Conventionally-reared D. melanogaster were allowed to oviposit on grape juice agar for 24 h (100 g/L Glucose, 100 g/L Yeast, 10 g/L agar, 10% grape juice concentrate). First instar larvae were then collected in PBS and transferred to 60 mm petri plates containing 1.2% agar in distilled water, 15 larvae per plate. About 100 mg of yeast cells suspended in 100 µl of 20% glucose were added as the source of nutrition. These included cells of S. bacillaris or S. bombicola ATCC22214 from overnight cultures, or dead lyophilized brewer’s yeast. Plates were covered and incubated at 25 °C for 6 days, then the proportion of larvae surviving to pupation was determined. To test the influence of spent culture supernatants, overnight cultures of each yeast were centrifuged at 14,000×  g for 30 s. The supernatant was transferred to a microcentrifuge tube filter column with a 0.45 µm cellulose acetate filter (Costar #8163) and centrifuged again. The filtered supernatant was used to resuspend dead brewer’s yeast, which was then fed to larvae.

Yeast survival in Drosophila larvae

Larvae were collected, transferred to petri plates, and fed suspensions of live yeast suspended in 20% glucose as described above: 15 larvae per plate. After 60 min of feeding, plates were flooded with sterile PBS and the larvae were transferred to a fresh agar plate using a clean paintbrush. Larvae were washed in 10% Bleach for two minutes, then rinsed twice in sterile PBS. Using a clean paintbrush, individual larvae were transferred to microcentrifuge tubes with 100 µl of sterile PBS and ∼100 µl of autoclaved ceramic beads (1.4 mm diameter). Larvae were homogenized with a vortex mixer for 30 s, then the homogenate was diluted and spread plated to determine the viable count of yeast in each larva.

Microscopy

Larvae from three independent yeast survival experiments were imaged alive under brightfield microscopy at 200×  and 630×  magnification on a Zeiss LSM 700 inverted microscope. Representative images were captured using Zeiss Blue software, and cropped to show areas of interest.

Statistics

Data were analyzed in R Software for Statistical Computing, version 3.3.1. Mann–Whitney pairwise tests were made with the wilcox.test function, and P values were adjusted for multiple comparisons by the Bonferroni correction. Development data were analyzed using the Survival, coxme, and multcomp packages as in Newell & Douglas (2014). Briefly, a cox mixed-effects model was applied to the survival functions describing the effect of microbial treatment on development time, and experimental replicate was included as a random effect in the model to account for any “block” variation among experiments. The glht function was used to apply Tukey’s Contrasts test to the results and P values were adjusted for multiple comparisons using the single-step method.

Results

Trapping of D. suzukii in Oswego County

Traps were monitored continuously from June 12 to July 31, 2017. A total of 45 D. suzukii individuals were captured, while 539 individuals from other species of Drosophila were also recovered. These data confirmed the presence of Drosophila suzukii in Oswego County, New York. Twice the amount of D. suzukii were captured in the orchard (30) as compared to adjacent land at the Rice Creek Field Station (15), which includes wooded and open areas. All Drosophila species, including D. suzukii, were caught most frequently in mid to late July (Fig. 1).

Figure 1 Drosophilidae captured during the survey period.

Drosophila sukukii (blue dots) were visually distinguished from other Drosophilidae (orange dots) and enumerated at each time point. Note: the y-axis is split to show that 200 flies were captured on July 24th.

Isolation of microbiota from D. suzukii

The traps employed a mesh covering that prevented flies from contacting the dough bait once inside the trap. D. suzukii individuals were chosen for microbiome analysis only if they were alive at the time of capture. Whole flies were individually homogenized and spread plated on selective media for bacteria or yeasts. The results showed that colony forming units (CFU) per fly varied across three orders of magnitude in D. suzukii (Fig. S1). Both yeasts and bacteria were recovered from every individual sampled. Bacterial density was slightly higher than yeast density in our dataset (Mann–Whitney, P < 0.05).

Identification of microorganisms associated with D. suzukii

Preliminary taxonomic identification of bacteria isolated from D. suzukii was performed by PCR amplification of the full-length 16S rRNA gene and automated Sanger sequencing of the V1–V3, and/or V7–V9 variable regions (Table 1). Among the 21 bacteria we were able to identify from D. suzukii, seven were from the Acetobacteracea family and eight from the Enterobacteriaceae. Outside of those groups, Pseudomonas was the most common genus, with four isolates. Our results are comparable to similar surveys of bacteria associated with D. suzukii (Vacchini et al., 2017; Martinez-Sañudo et al., 2018).

Table 1 Bacteria isolated from D. suzukii.

Isolate	Top BLAST hit accession	Alignmentlength	%ID	16S regions	
Acetobacter malorum OSW_437_dd	NR_113553.1	846	99.65%	V1–V3	
Acetobacter persici OSW_443_jj	NR_113552.1	398	98.49%	V1–V3	
Asaia lannensis OSW_426_N	NR_114144.1	1,016	99.41%	V7-V9	
Asaia siamensis OSW_449_pp	NR_113845.1	501	99.40%	V1–V3	
Comamonas testosteroni OSW_413_10	NR_113709.1	951	99.79%	V7–V9	
Enterobacter sp. OSW_435_bb	NR_146667.2	453	94.48%	V1–V3	
Erwinia aphidicola OSW_423_J	NR_104724.1	526	98.10%	V1–V3	
Erwinia sp. OSW_405_5	NR_118431.1	367	98.37%	V1–V3	
Erwinia rhapontici OSW_434_aa	NR_118858.1	755	97.09%	V1–V3	
Gluconobacter cerinus OSW_446_mm	NR_118192.1	906	99.01%	V7–V9	
Gluconobacter frateurii OSW_444_kk	NR_118193.1	901	99.00%	V1–V3	
Gluconobacter japonicus OSW_424_L	NR_118638.1	980	99.69%	V7–V9	
Leuconostoc sp. OSW_442_ii	NR_109004.1	430	97.44%	V1–V3	
Pseudomonas endophytica OSW_427_P	NR_136473.1	507	99.21%	V1–V3	
Pseudomonas endophytica OSW_436_cc	NR_136473.1	599	99.50%	V1–V3	
Pseudomonas putida OSW_411_8	NR_113651.1	872	98.97%	V1–V3	
Pseudomonas coleopterorum OSW_422_I	NR_137215.1	689	99.42%	V1–V3	
Rosenbergiella epipactidis OSW_412_k	NR_126303.1	396	96.46%	V1–V3	
Rosenbergiella sp. OSW_404_o	NR_104901.1	820	99.39%	V1–V3	
Shigella boydii OSW_438_ee	NR_126303.1	554	99.64%	V1–V3	
Tatumella sp. OSW_445_ll	NR_116799.1	578	94.29%	V1–V3	

Preliminary taxonomic identification of yeasts isolated from D. suzukii was conducted by sequencing the ITS regions of the rRNA locus. Five genera were identified among the 16 isolates from which sequences were obtained. The most prevalent genus was Hanseniaspora (Table 2), consistent with previous surveys of fungi associated with D. suzukii (Hamby et al., 2012; Lewis et al., 2019). Three yeasts isolated from D. melanogaster caught in our traps were also sequenced and included in subsequent experiments.

Table 2 Yeast isolated from D. suzukii and D. melanogaster.

Isolate	Source	Top BLAST hit accession	Alignment length	%ID	
Candida railenensis OSW_409_6	D. suzukii	HQ438312.1	555	99.82%	
Candida railenensis OSW_417_D	D. suzukii	HQ438308.1	570	99.82%	
Candida railenensis OSW_455_vv	D. suzukii	HQ438312.1	558	99.46%	
Hanseniaspora sp. OSW_452_ss	D. suzukii	KU350327.1	163	95.71%	
Hanseniaspora uvarum OSW_428_Q	D. suzukii	KY103571.1	519	100.00%	
Hanseniaspora uvarum OSW_429_R	D. suzukii	KY103552.1	522	100.00%	
Hanseniaspora uvarum OSW_431_T	D. suzukii	KY103571.1	523	98.85%	
Hanseniaspora uvarum OSW_416_C	D. suzukii	MG250501.1	689	99.71%	
Hanseniaspora uvarum OSW_419_F	D. suzukii	MK352062.1	475	96.00%	
Hanseniaspora vineae OSW_430_S	D. suzukii	KY103581.1	648	99.85%	
Metschnikowia sp. OSW_457_xx	D. suzukii	KM243742.1	319	99.69%	
Metschnikowia sp. OSW_451_rr	D. suzukii	KF690368.1	266	95.86%	
Metschnikowia sp. OSW_456_ww	D. suzukii	KF690368.1	266	95.86%	
Saccharomyces sp. OSW_433_V	D. suzukii	KX905283.1	347	90.20%	
Starmerella bacillaris OSW_450_qq	D. suzukii	KU950242.1	401	99.00%	
Starmerella bacillaris OSW_454_uu	D. suzukii	MK352049.1	403	100.00%	
Candida tropicalis OSW_414_B	D. melanogaster	MK752673.1	449	99.78%	
Meyerozyma guilliermondii OSW_453_tt	D. melanogaster	MK547245.1	491	99.39%	
Pichia kudriavzevii OSW_421_H	D. melanogaster	MK894151.1	444	100.00%	

Impact of bacteria on larval development in Drosophila

We monitored the development time of D. melanogaster from the embryo to the pupal stage under mono-associated gnotobiotic conditions. This experiment focused mainly on isolates from D. suzukii, though a few yeast isolates from D. melanogaster were also included. First, the impact of individual species of bacteria were compared. Consistent with prior studies, mono-association with some Acetobacter, Gluconobacter, and Pseudomonas species accelerated larval development relative to axenic controls (Fig. 2; Table 3). The effects of other species tested were mixed, with Rosenburgiella sp. producing the most rapid development, and C. testosteroni the slowest. However, these differences were not significantly different from axenic conditions when correcting for multiple comparisons (Table 3). All of the microorganisms we tested were able to proliferate in the Drosophila vials during development experiments except P. coleopterorum, which was not recovered from diet samples (Fig. S2).

Figure 2 Larval development of gnotobiotic D. melanogaster mono-associated with bacteria.

Kaplan Meier curves depict the probability of pupariation at each time point after egg deposition based on observations of gnotobiotic Drosophila mono-associated with the bacteria indicated in the inset legends. Each line in the plot corresponds to the aggregate data for a single microbial treatment. For each treatment n = 96 to 811 individuals (median 159) from two to five independent experiments. The black line indicates development of axenic larvae. Data are grouped as follows: (A) Pseudomonas species, (B) acetic acid bacteria, (C) Enterobacteria and C. testosteroni. Table 3 summarizes statistics comparing each treatment to axenic conditions.

Table 3 Cox survival model statistics comparing development of gnotobiotic D. melanogaster.

Treatments compared	Estimate	SE	z value	P value	
A. malorum dd - axenic	0.587	0.090	6.489	<0.01	
A. persici jj - axenic	0.614	0.074	8.328	<0.01	
Asaia lannensis N - axenic	0.189	0.096	1.974	0.776	
Asaia siamensis pp - axenic	0.026	0.104	0.252	1.000	
C. testosteroni 10 - axenic	−0.157	0.062	−2.544	0.363	
E. rhapontici aa - axenic	0.316	0.152	2.079	0.706	
Enterobacter sp. bb - axenic	−0.118	0.139	−0.853	1.000	
G. frateurii kk - axenic	0.296	0.071	4.153	<0.01	
G. japonicus L - axenic	0.578	0.105	5.502	<0.01	
P. coleopterorum i - axenic	−0.194	0.104	−1.869	0.838	
P. endophytica cc - axenic	1.054	0.100	10.509	<0.01	
P. endophytica P - axenic	0.736	0.102	7.195	<0.01	
R. epipactidis o - axenic	0.437	0.145	3.02	0.124	
Tautumella sp. ll - axenic	0.071	0.135	0.524	1.000	
C. railensis 6 - axenic	1.143	0.056	20.228	<0.01	
C. railensis D - axenic	1.468	0.087	16.859	<0.01	
C. tropicalis B - axenic	1.108	0.141	7.862	<0.01	
H. uvarum C - axenic	0.946	0.098	9.653	<0.01	
H. uvarum F - axenic	0.992	0.118	8.41	<0.01	
H. uvarum R - axenic	1.097	0.108	10.148	<0.01	
H. vineae S - axenic	1.215	0.089	13.706	<0.01	
Hanseniaspora sp. ss - axenic	1.063	0.059	18.162	<0.01	
Metschnikowia sp. rr - axenic	1.122	0.075	14.876	<0.01	
Metschnikowia sp. ww - axenic	0.953	0.086	11.055	<0.01	
Metschnikowia sp. xx - axenic	1.078	0.108	9.996	<0.01	
Meyerozyma tt - axenic	1.171	0.103	11.345	<0.01	
P. kudriavzevii - axenic	0.975	0.103	9.469	<0.01	
S. bacillaris qq - axenic	−0.358	0.058	−6.192	<0.01	
S. bacillaris uu - axenic	−0.408	0.063	−6.441	<0.01	
Saccharomyces sp. V - axenic	0.466	0.070	6.636	<0.01	

Impact of yeast on larval development in Drosophila

Next, we examined the impact of individual species of yeast. Nearly every isolate we tested accelerated larval development relative to axenic conditions (Table 3), with Candida tropicalis (a D. melanogaster isolate) and Hanseniaspora species (all D. suzukii isolates) producing the earliest pupariation times (Fig. 3). Starmerella bacillaris (syn. Candida zemplinina, (Masneuf-Pomarede et al., 2015)) was unique among yeasts in that it significantly slowed larval development relative to axenic conditions. This result was observed with two S. bacillaris isolates from different D. suzukii individuals from different sampling sites.

Figure 3 Larval development of gnotobiotic D. melanogaster mono-associated with yeast.

Kaplan Meier curves depict the probability of pupariation at each time point after egg deposition based on observations of gnotobiotic Drosophila mono-associated with the yeasts indicated in the inset legends. Each line in the plot corresponds to the aggregate data for a single microbial treatment. For each treatment n = 108 to 811 individuals (median 204) from two to five independent experiments. The black line indicates development of axenic larvae. Yeasts isolated from wild D. melanogaster are indicated with an asterisk. Data are grouped as follows: (A) Hanseniaspora species, (B) Candida species, (C) Metschnikowia as well as other species, and (D) Starmerella bacillaris. Table 3 summarizes statistics comparing each treatment to axenic conditions.

To see if this was a general property shared among yeasts of the Starmerella genus, S. bombicola ATCC22214, an isolate from bumblebee honey, was tested for its effect on Drosophila development. Gnotobiotic larvae mono-associated with S. bombicola develop significantly faster than axenic larvae or larvae reared with S. bacillaris (Fig. S3). This suggests that prolonging development is not a general property of Starmerella yeasts. Instead, S. bombicola resembles the other species of yeasts characterized in this study in that it can accelerate larval development.

Starmerella bacillaris cells do not nourish larvae

We tested two hypotheses that could explain how S. bacillaris prolongs Drosophila development: (a) S. bacillaris produces a soluble product that inhibits larval growth, or (b) S. bacillaris cells do not serve as a good source of nutrition for larvae. First, we transferred D. melanogaster larvae to non-nutritive agar plates and added dead brewer’s yeast resuspended in either fresh or spent YPD medium as the source of nutrition. Spent YPD medium from yeast cultures was collected and sterilized by centrifugation and filtration. There was not a significant difference in larval survival among the treatments tested (Fig. 4A; Mann–Whitney, P > 0.05), indicating that S. bacillaris supernatant did not negatively affect larval survival relative to fresh YPD or supernatant from S. bombicola cultures.

Figure 4 Starmerella bacillaris cells do not nourish D. melanogaster larvae.

(A) The survival of larvae to pupariation was monitored on non-nutritive agar supplemented with suspensions of dead brewer’s yeast in fresh YPD medium or spent culture supernatant (supe) from S. bacillaris (Sbac) or S. bombicola (Sbom). Proportion surviving was not significantly different across treatments (Mann–Whitney, P > 0.05; n = 11 replicates of 15 larvae each across three independent experiments). (B) Larval survival was monitored as in (A) but with dead brewer’s yeast (BY) or live yeast cells suspended in 20% glucose as the source of nutrition. Survival was significantly lower with S. bacillaris OSW_450_qq (Sbac qq) compared to S. bombicola (Sbom) or BY (Mann–Whitney, P < 0.001; n = 11 from 3 independent experiments).

To test whether S. bacillaris cells could serve as a source of nutrition for larvae, live yeast cells (or dead brewer’s yeast) were resuspended in a 20% glucose solution and added to non-nutritive agar plates as the only source of food. Under these conditions, a median of 45% of larvae survived to pupation when dead brewer’s yeast was provided as food (Fig. 4B). When live Starmerella bombicola ATCC22214 cells were provided as food, around 20% of larvae survived to pupation. By contrast, live S. bacillaris cells did not support larval survival under these conditions, as only 4 out of 165 larvae tested survived to pupation (Fig. 4B). These results are consistent with the hypothesis that S. bacillaris cells are a poor source of nutrition for developing larvae.

More viable Starmerella bacillaris cells in larvae compared to other yeasts

Given the evidence that S. bacillaris cannot serve as a source of nutrition for Drosophila larvae, we investigated whether S. bacillaris cells are ingested by larvae and if they differ from other yeasts in their ability to survive consumption by larvae. To observe yeast ingestion and measure the number of viable yeast cells inside of larvae, L1 larvae were fed dense suspensions of yeast in 20% glucose for 1 h, then surface sterilized and washed with sterile PBS before homogenization and plating. Larvae fed continuously regardless of which yeast was provided based on microscopic observation. However, the number of viable yeast cells per larva was significantly higher for S. bacillaris strains compared to S. bombicola or H. uvarum—two yeasts that support rapid larval development (Fig. 5; Mann–Whitney, P < 0.001). These results suggest that Drosophila larvae may not digest S. bacillaris to the same extent as other yeasts.

Figure 5 Viable cell density of yeasts inside of D. melanogaster larvae.

L1 larvae were fed suspensions of the yeasts indicated, then surface sterilized, homogenized and plated to determine colony forming units (CFU) per larva. In each whisker box plot, the box delineates the first and third quartiles, the dark line is the median, and the whiskers show the range (minus outliers, which appear as circles). S. bacillaris OSW_450_qq (Sbac qq) and S. bacillaris OSW_455_uu (Sbac uu) both had a higher viable cell density in larvae than S. bombicola (Sbom) or H. uvarum OSW_429_R (Huva) (Mann-Whitney, P < 0.001; n = 31 to 39 from 3 to 4 independent experiments).

To visualize whether larval digestion of S. bacillaris differs from yeasts that support development, we utilized brightfield microscopy to observe live larvae from the feeding experiment above (after washing). Frass excreted from larvae fed H. uvarum was heterogenous; some whole yeast cells were visible, but they were surrounded by debris and particles of various sizes (Fig. 6A). Surprisingly, frass from larvae fed S. bacillaris was a nearly uniform mass of whole yeast cells (Fig. 6B). While frass from larvae fed H. uvarum dispersed easily, frass containing S. bacillaris was excreted in long, compact trails that did not disperse despite the continued movement of larvae (Fig. 6C). Closer observation of the frass trails revealed that they consisted of cells densely packed in a clear sheath (Fig. 6D). These observations were corroborated in three independent experiments.

Figure 6 Starmerella bacillaris cells are intact after passage through larvae.

L1 larvae were fed suspensions of H. uvarum OSW_429_R (A) or S. bacillaris OSW_450_qq (B–D), washed, then imaged live under brightfield microscopy. Frass excreted from larvae is pictured in A and B. A trail of frass containing S. bacillaris is indicated by the arrow in C, and a clear sheath structure surrounding the trail is indicated by the triangle in D.

Discussion

We investigated how microbes isolated from D. suzukii could impact D. melanogaster larval development to learn more about potential interactions between these species. Our results show significant impacts on D. melanogaster, some positive and some negative. We found that nearly all yeasts isolated from field-caught flies accelerated larval development under gnotobiotic conditions, with the exception of Starmerella bacillaris. Here we discuss this intriguing result, possible mechanisms behind the effects we observed on development, the findings of our survey, and the broader implications of interactions between these microbes and Drosophila.

The effects of yeasts on D. melanogaster development

Two independent isolates of S. bacillaris (syn. Candida zemplinina) antagonized the growth of larvae (Fig. 3D) and appeared to be indigestible to D. melanogaster. This yeast species is commonly found on grapes and in wine (Masneuf-Pomarede et al., 2015), and has been identified in a number of surveys of yeasts associated with Drosophila (Hamby et al., 2012; Stamps et al., 2012; Lewis et al., 2019), so it is plausible that these interactions occur in nature. Prior studies have shown D. melanogaster and D. suzukii require microorganisms to complete larval development on low-protein diets (Wong, Dobson & Douglas, 2014; Bing et al., 2018). In contrast, experiments utilizing nutrient-rich conditions (like the diet used here), have shown more subtle effects of the microbiota on developmental rate, with some microbial taxa promoting development and others not (Newell et al., 2014; Chaston, Newell & Douglas, 2014). As S. bacillaris is the first microbe we have observed to slow development on this diet relative to axenic conditions, we hypothesize that it competes with D. melanogaster for nutrients, effectively lowering the quality of the diet. The clear, sheath-like structure surrounding S. bacillaris cells in larval frass (Fig. 6D) may protect them from digestion, though this is only speculation. It may be composed of proteins or carbohydrates produced by the yeast, or potentially by the larvae themselves.

All of the other yeast isolates we tested significantly accelerated the development of larval D. melanogaster. This suggests the possibility that a wide diversity of yeasts could accelerate development via a common mechanism –for example, by altering protein/carbohydrate ratios in the diet (Wong, Dobson & Douglas, 2014), or liberating amino acids (Yamada et al., 2015). Alternatively, mechanisms unique to certain yeast taxa may result in similar outcomes for Drosophila development.

Bacteria and D. melanogaster development

Bacteria isolated from D. suzukii had varied effects on larval development time (Fig. 2). Each of the isolates that accelerated development in this study belong to genera that have previously been shown to do so (Chaston, Newell & Douglas, 2014). A metagenome-wide analysis by Chaston et al. found that oxidative metabolism genes of the microbiota are significantly associated with faster development in gnotobiotic flies, especially dehydrogenases that employ the cofactor pyrroloquinoline quinone (PQQ) (Chaston, Newell & Douglas, 2014). Interestingly, the two Asaia species tested here did not significantly alter development time relative to axenic conditions. Asaia are AAB commonly isolated from insects (Crotti et al., 2010) but may not possess the PQQ-dependent alcohol dehydrogenase associated with promotion of larval development (Ano et al., 2008; Shin et al., 2011).

To our knowledge, this study is the first to examine the impacts of Enterobacteriaceae isolated from field-caught flies on Drosophila development. Results were mixed, with none of the species significantly altering development of gnotobiotic larvae relative to the axenic control. This was true even for an isolate of Tatumella, an organism previously identified as dominant in cherries infested with D. suzukii (Chandler et al., 2014).

The D. suzukii population surveyed

Our survey focused on two small areas in relatively close proximity: one an orchard and the other a partially wooded ecological research station. The orchard was not experiencing a D. suzukii infestation, and neither site had abundant oviposition sites (ripening fruit) for these files in the immediate vicinity of our traps. Given the duration of our survey and the relatively low abundance of D. suzukii, we presume the individuals we caught likely fed on a range of food sources and may not represent one population. A number of studies have highlighted the importance of forests as a habitat for D. suzukii and one recently showed that proximity to forests increased trapping of D. suzukii in cherry orchards (Hennig & Mazzi, 2018). It should also be noted that trapping bias has been observed in D. suzukii; virgin females, protein starved females, and males tend to prefer vinegar-based baits like the ones used in this study, while ovipositing females are more attracted to fruit volatiles (Clymans et al., 2019). Therefore, it is possible our microbial isolates are skewed toward a subset of the D. suzukii population(s).

Microorganisms isolated from D. suzukii

Despite the small scale of our survey, the isolates we obtained are typical of those found in previous culture-based and culture-independent studies of D. suzukii microbiota. Pioneering work by Hamby et al. characterized yeasts associated with D. suzukii, finding that Hanseniaspora uvarum was the predominant species isolated (Hamby et al., 2012). Our results agree with that conclusion, though it should be noted that there may be a cultivation bias for H. uvarum due to its rapid growth rate and ability to outcompete other yeasts (Lewis et al., 2019). Surveys of bacteria published to date found Acetobacteraceae and Enterobacteriaceae to be prominent constituents of the D. suzukii microbiota (Chandler et al., 2014; Vacchini et al., 2017; Rombaut et al., 2017; Martinez-Sañudo et al., 2018).

The impacts of AAB on D. melanogaster biology have been well studied: they can influence development time, fecundity, and nutrition among other traits (Newell & Douglas, 2014; Gould et al., 2018; Walters et al., 2018; Sannino et al., 2018). Some of these findings have been extended to D. suzukii as well (Bing et al., 2018). Vacchini et al. observed a high prevalence of AAB in wild-caught D. suzukii, and found that changes in the microbiota of adults upon a shift from fruit-based to sugar-based diets primarily occurs in AAB species composition (Vacchini et al., 2017). Comparatively little is known about the Enterobacteriaceae associated with vinegar flies, despite their frequent identification in microbiota surveys. Interestingly, a recent survey by Martinez-Sanudo et al. found a higher abundance and diversity of Enterobacteriaceae in D. suzukii caught in newly colonized regions (Martinez-Sañudo et al., 2018). Whether this shift is indicative of differences in the diet utilized by the flies in different locations or reflective of other adaptations to a new environment is unknown.

Broader implications

For dietary microbes like yeasts, there is likely a tradeoff between the benefits of dispersal and the risk of digestion in the host (Garcia & Gerardo, 2014; Broderick, 2016; Inamine et al., 2018). It appears that S. bacillaris could maximally benefit from being consumed and dispersed by D. melanogaster by avoiding death in the gut. This would shift the usually mutually beneficial relationship between flies and yeast to one in which the yeast benefits at the expense of the fly. We should note that our experiments only examined S. bacillaris survival in larvae, and it is unknown whether adult D. melanogaster or any stage of D. suzukii would give similar results. Interactions between microbial species are also likely to drive changes in the microbial communities found at feeding and oviposition sites of D. suzukii and D. melanogaster (Fischer et al., 2017; Álvarez Pérez, Lievens & Fukami, 2019). More research examining these interactions is needed, including the dynamic role Drosophila larvae can play in modifying the microbial ecology of their substrates (Stamps et al., 2012; Lewis et al., 2019), in order to gain a broader understanding of the processes that drive microbiota assembly in this system (Adair & Douglas, 2017).

While most studies have found beneficial relationships between individual yeasts and Drosophila in laboratory studies, a few have noted a disconnect between the attractiveness of yeasts to ovipositing females and the effects of those yeasts on offspring performance (Anagnostou, Dorsch & Rohlfs, 2010; Anagnostou, LeGrand & Rohlfs, 2010; Buser et al., 2014; Hoang, Kopp & Chandler, 2015; Bellutti et al., 2018). This has led to the suggestion that yeast volatiles may not always be a true signal of the quality of a substrate for oviposition—i.e., the fitness benefit to developing larvae. However, differences in diet and inconsistencies in controlling for other microbiota (i.e., bacteria) across these studies limit the utility of comparing results. Future studies should control for these variables to seek a more comprehensive view of the Drosophila microbiota that includes both bacterial and yeast constituents and utilizes recently isolated microbial strains that have not adapted to the lab environment.

Limitations of this study

There are two major caveats to acknowledge in the interpretation of our development data. One is that our experiments were performed on a nutrient-rich laboratory diet rather than fruit-based substrates. A benefit of our laboratory diet is that gnotobiotic Drosophila do not depend on the microorganisms for survival to pupation. In fact, axenic larvae develop to adulthood in 10–11 days on the diet, which is comparable to conventionally reared flies in many studies. This means differences observed may reflect more subtle influences of microbes on the timing of development. However, in future studies, a holidic diet in which the contents can be precisely manipulated would be more useful for determining which nutrients S. bacillaris may compete for with larvae (Piper et al., 2014). The second caveat is that we utilized single-species gnotobiotic associations, and thus did not examine how interactions between microbes would impact the host. Interspecies interactions are a key element of microbiota function in D. melanogaster (Newell & Douglas, 2014; Gould et al., 2018; Sommer & Newell, 2019). Examining how the Enterobacteriaceae or yeasts we isolated interact with other, better-studied members of the Drosophila microbiota is a ripe area for further investigation.

Conclusions

We conclude that the microbiota of D. suzukii can significantly alter the development time of D. melanogaster larvae. Yeasts accelerate development, aside from S. bacillaris which significantly prolongs the larval period. Future work will test the hypothesis that S. bacillaris competes with Drosophila for nutrients, and investigate the mechanism by which S. bacillaris may survive passage through the larval gut. Additional research into interactions between microbial species isolated in this study will further elucidate how the microbiota of D. suzukii influence D. melanogaster. More broadly, we view the microbiota as an important axis in the interactions between Drosophila species, and as a valuable tool for understanding their ecology.

Supplemental Information

Supplemental Information 1 Raw data for all non-image figures

All the raw data for the article excluding the image files for Fig. 6. Each tab in the XLSX file corresponds to a figure or figure panel.

Click here for additional data file.

Figure S1 Quantification of microorganisms in D. suzukii

Field-caught flies were homogenized and plated on selective media to culture bacteria and yeasts separately. In each whisker box plot, the box delineates the first and third quartiles, the dark line is the median, and the whiskers show the range (minus outliers, which appear as circles). Bacterial colony forming units (CFU) per fly were significantly higher than that for yeasts (Mann–Whitney, P < 0.05, n = 15).

Click here for additional data file.

Figure S2 Viable cell density of microorganisms in the diet of mono-associated D. melanogaster

Bacteria (A) and yeasts (B) were quantified by serial dilution and plating on YPD. In each whisker box plot, the box delineates the first and third quartiles, the dark line is the median, and the whiskers show the range. *no CFU were recovered for P. coleoperorum.

Click here for additional data file.

Figure S3 Development of D. melanogaster larvae mono-associated with Starmerella bombicola

Kaplan–Meier plots depict the survival functions estimating the probability of pupariation at each time point after egg deposition based on observations of gnotobiotic Drosophila mono-associated with the yeast indicated in the inset legend. Each treatment was significantly different from the others in a Cox mixed-effects model, P < 0.01, n = 221 to 351 from two independent experiments.

Click here for additional data file.

Figure S6A Raw microscopic image of larval frass

Click here for additional data file.

Figure S6B Raw image of larval frass pictured in Fig. 6B

Click here for additional data file.

Figure S6C Raw image of larval frass trail from Fig. 6B

Click here for additional data file.

Figure S6D Raw image of larval frass trail Fig. 6D

Click here for additional data file.

We thank Luiza Nawrot and Andrew Sommer for assistance with laboratory work. We thank the Gfeller family for access to their land.

Additional Information and Declarations

Competing Interests

Author Contributions

DNA Deposition

Data Availability

The authors declare there are no competing interests.

Gabrielle M. Solomon and Peter D. Newell conceived and designed the experiments, performed the experiments, analyzed the data, contributed reagents/materials/analysis tools, prepared figures and/or tables, authored or reviewed drafts of the paper, approved the final draft.

Hiruni Dodangoda, Tylea McCarthy-Walker and Rita Ntim-Gyakari performed the experiments, prepared figures and/or tables, approved the final draft.

The following information was supplied regarding the deposition of DNA sequences:

The sequences are available in GenBank: bacterial sequences, MN197709 –MN197729; yeast sequences, MN209205 –MN209223.

The following information was supplied regarding data availability:

The raw data is available in the Supplementary Files.

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
