# Peer review of "The microbiota of Drosophila suzukii influences the larval development of Drosophila melanogaster"

_PeerJ, doi:10.7717/peerj.8097_

## Round 0.1 · original submission · Minor Revisions

The manusript has been evaluated by three different reviewers, who all agreed that this is an excellent and well-written study, which was my impression as well. However, they suggested a number of revisions that will improve the manuscript further. I believe that it will be acceptable for publication after they have been implemented.

Reviewer 1 ·

Basic reporting

The paper is very well written. It is concise, well planned and executed. The key findings and their implications in Drosophila ecology through the microbiota comes through.

Experimental design

In the Introduction section, line 59 gives the impression that the interaction of D. suzukii with its microbiome is the main aim. The actual aim of the project is not introduced until line 75. The main aim of the project should be introduced early on in the Introduction and story line should be build up from there.

Similarly, the abstract can be restructured to highlight the ecological importance of D. suzukii and D. melanogaster interaction, and why the microbiome is a good tool to study that.

In the methods section, the statistical tests in the statistical packages are not mentioned.
Only the proportion of larvae is given in Figure 5. Please add the number of larvae used for feeding experiment in methods section.

Validity of the findings

The results section "Impact of microbiota on larval development in Drosophila" can be broken down into two sub-headings, namely bacteria and yeast.

Line 223- separate the results of yeasts isolated from D. suzukii and D. melanogaster.

The discussion section can be restructured. For example, the effects of bacteria on larval development is introduced in line 290 but discussed in line 339. Finish one sub-topic and then move to another one. Also add the figure number in parenthesis in lines discussing the result.

Mention the importance of microbiome as a tool and aim of studying the interactions between Drosohila species in Conclusion section to highlight the importance of the study.

Additional comments

The paper is very well structured and engaging. The methods are clearly explained, results are good quality with relevant figures. The results are well discussed with citations from recent literature. The study is sound and exciting.

·

Basic reporting

Line 204-205: The statement “Monitoring took place a minimum of every four days, and fresh bait was introduced at the same interval” does not correspond to the dates provided in the raw data excel sheet supplement.

Line 213 (beginning with ‘as’) to line 221 (Douglas, 2014) does not belong in the results section and should be moved to the intro or discussion.

Figure 2: It is unclear why figure 2 is a main figure, as opposed to a supplemental figure, as there is not discussion of the significance of bacteria having a higher CFU count relative to yeast anywhere in the paper. Even more, figure 2 is not even mentioned in the text of the paper, and the only mention of such findings are in lines 195-196. It is recommended that this figure be moved to supplemental files.

Figures 3 and 4: The plots of figures 3 and 4 are quite difficult to look at and read. First, the black line denoting controls is very difficult to see, particularly in graphs with blue labels, as well as in figure 3 overall. It is suggested to take a step back and think of an alternative way to present these findings that may be easier for the reader to take in. At minimum, the control line should be better defined.

Experimental design

Although the study design appears well thought out, the flow is difficult to follow and is not easily grasped unless one reads the entire paper repeatedly.
• To help readers to more quickly comprehend the study design, a short summary (or even better, a figure) describing an overview of the methods in the beginning of the methods section would help the reader to better follow the design.
• You often leave steps and justifications out of your methods (see specific examples below), and then talk about them in the results. Specific examples of incomplete method descriptions are:
• Lines 108-118: Were homogenization beads reused and cleaned between homogenizations? If so, using what protocol?
• Lines 117: How were colonies of interest determined? More specific details needed about how you identified the different yeast and bacteria species and why you selected them. These answers can eventually be found in lines 198-200 of the methods, where you state that preference was given to most prevalent types. This should be first stated in the methods section. Additionally, why was preference given to prevalence, particularly in cultured conditions?
• Lines 123-124: What specific regions of rRNA were sequenced? You specify bacteria in results (lines 201-202), but this should be stated in the methods.
• Lines 123-132: Authors should provide citations for choosing the hypervariable regions selected for 16s and ITS.
• Lines 136-137: Although you specify that Sanger sequencing was used in the results (lines 200-202), this should be clearly stated in the methods section. Otherwise, the reader is left wondering the method used for sequencing.
• Lines 137-138: How was the highest quality sequence chosen?
• Lines 142-143: What was the source of the Canton S D. melanogaster flies?

Lines 77-78: Authors should be more specific in this sentence to describe development time from dechorionated embryo to pupation, as it is not immediately obvious.

Lines 108-111. It should be noted whether flies of interest included both males and females.

Lines 108-177/Supplement: How many isolates of each microbial species were tested on larval development time? You note that Starmerella bacillaris induced delayed development from two separate isolates, and it is clear that multiple experiments have been conducted based on the text within your main figures. However, the specifics of replication should be included in the Supplementary Raw Data file as well as in the methods section.

Validity of the findings

Generally, the overall findings, discussion, and conclusion appear valid, with one small addition suggested for the discussion (see below). Additionally, there is some missing information from the supplementary raw data file, which are needed for external investigators to be able to independently reassess the findings (see below).

The supplementary Raw Data file
• The Supplementary Raw Data File as a whole would benefit by having a key for each tab. In particular, it is impossible to reanalyze the data or viably recreate any figures with the information provided for figures 3 and 4, particularly with regards to what Treatment letters denote.
• Figure 5 data does not specify the three independent treatments, leaving it impossible to determine the validity of the experimental design.
• Figure 6 also does not denote what was measured for each of the three or four independent experiments conducted.

Drosophila melanogaster larvae showed delayed development time in response to Starmerella bacillaris exposure, which you support with additional work demonstrating indigestibility of S. bacillaris in the larval gut. Yet, you also state that the flies reared in this experiment do not depend on the mono-associated microorganisms for sustenance, as experiments were performed on a nutrient-rich laboratory diet rather than fruit-based substrates (lines 355-358). Given this, it would be beneficial to discuss this discrepancy, even if briefly, in the discussion section.

Additional comments

The primary objective of this paper was examine the influence of mono-associated yeast and bacteria species originally derived from Drosophila suzukii hosts on Drosophila melanogaster development time (i.e. from first instar to pupation). Additionally, the authors identified the presence of D. suzukii in Oswego County, NY. The noteworthy finding demonstrated that the mono-associated yeast species, S. bacillaris (obtained from D. suzukii) significantly delayed D. melanogaster larval development, which was attributed to indigestibility of S. bacillaris by the larval gut of D. melanogaster, thus providing poor nutritional value to larva.

Overall, the experiment appears to be well thought out and appropriately executed. I believe this paper is a strong candidate for publication, with only a few concerns. Primarily, the methods are quite difficult to follow as written (see “Experimental Design” section), as much of the split between the methods and results sections, and the supporting documentation is missing a few pieces of data in order for the experiment to be reproducible by another investigator (see “Validity of Findings” section). The other comments are very minor in comparison and include additional details/specifics needed (see “Basic Reporting” and “Experimental Design” sections), and grammatical and typo issues (see below).

Grammatically confusing suggestions:
• Line 42: The word ‘consummate’ is not clear here in the context of the remainder of the sentence. Clarify why being a consummate model organism is important to studying host-microbe interactions or use a more appropriate adjective.
• Line 319-320: This sentence would benefit from an oxford comma following “protein starved females”, as it is confusing when reading whether it is all males or only protein starved males that prefer vinegar-based bates.

Typos:
• Line 119. The letter L in liquid is bolded.
• Line 146. Missing end parenthesis ).
• Line 147. Missing ‘at’… “Optical densities… were measured (at) 600nm.”

---

## Round 0.2 · accepted · Accept

The authors have thoroughly addressed the various minor comments made by the reviewers, including adding a more detailed statistical analysis section.